# Clinical Correlation of Bladder Electron Microscopic Characteristics in Patients with Detrusor Underactivity of Various Etiologies

**DOI:** 10.3390/biomedicines10051055

**Published:** 2022-05-03

**Authors:** Jia-Fong Jhang, Han-Chen Ho, Yuan-Hong Jiang, Yung-Hsiang Hsu, Hann-Chorng Kuo

**Affiliations:** 1Department of Urology, Hualien Tzu Chi Hospital, Buddhist Tzu Chi Medical Foundation, Tzu Chi University, Hualien 970, Taiwan; alur1984@tzuchi.com.tw (J.-F.J.); redeemerhd@tzuchi.com.tw (Y.-H.J.); 2Department of Anatomy, Tzu Chi University, Hualien 970, Taiwan; hcho@gms.tcu.edu.tw; 3Department of Pathology, Buddhist Tzu Chi General Hospital, Tzu Chi University, Hualien 970, Taiwan; yhhsu@tzuchi.com.tw

**Keywords:** urodynamic, urine retention, pathogenesis, treatment outcome

## Abstract

This study aimed to investigate the ultrastructural characteristics of the bladder of patients with detrusor underactivity (DU) of various etiologies. Twenty-five patients with DU and control subjects underwent urodynamic testing and transmission electron microscopic examination of bladder specimens. The epithelium, lamina propria, and muscle layers were analyzed separately. The DU bladders exhibited total epithelial denudation (52%). In the bladders with remaining epithelium, apical cell uroplakins (44.4%) and tight junction complexes (77.8%) were also noted. The lamina propria was characterized by loose extracellular connective tissue (48%) and a lack of nerve terminals (76%). Smooth muscle shrinkage and a loss of their regular spindle shape (91.6%) were also noted in the detrusor layer. Patients with DU with intact epithelial cell layers had significantly larger void volumes and maximal flow rates than those with mild or severe epithelial denudation. Patients with remaining nerve terminals in lamina propria had a stronger first sensation of filling and smaller residual urine volume than those without nerve terminals. The proportion of ultrastructural defects of the bladder was not significantly different among patients with DU of various etiologies and treatment outcomes. DU bladders were characterized by ultrastructural defects in the entire bladder, and the defects were correlated to clinical parameters.

## 1. Introduction

Detrusor underactivity (DU) is a urodynamic diagnosis that is defined as the reduction of urinary bladder contraction strength and/or duration, resulting in prolonged or failed bladder emptying [1]. Patients with DU are characterized as having a reduced urinary flow, increased post-void residual amounts, and often, decreased bladder sensation [1]. DU is a common urological disease among elderly patients, and a previous study reported that the prevalence of DU was 40.2% in elderly men and 13.3% in elderly women with lower urinary tract symptoms [2]. In contrast to overactive bladder and detrusor overactivity, DU has remained poorly researched, and the etiologies are largely unrecognized. The pathogenetic mechanism of DU involves complicated pathways. The possible etiologies of DU include central or peripheral nerve injury, bladder outlet obstruction, chronic ischemia, and aging [3]. While a previous pathogenetic study of DU focused on molecular and physiological changes in the bladder, the number of morphological and anatomic studies of DU bladders is limited. Evidence from animal and human studies of DU of various etiologies has shown histological characteristics of DU bladders, such as bladder wall thickening, inflammation, collagen deposition, and fibrosis [4,5]. However, the clinical significance of the pathological changes in DU bladders of various etiologies has not been established.

Electron microscopy (EM) has been widely utilized in the field of functional urology to investigate ultrastructural changes in the bladder. Previous animal DU studies that employed various models (e.g., ischemia or diabetic models) revealed ultrastructural changes in the bladder on EM, including decreased number of fusiform vesicles of umbrella cells, decreased number of caveolae, and muscle fascicle destruction [6,7]. Few studies have investigated ultrastructural defects in human DU bladders. Limited findings from human studies revealed swollen mitochondria and decreased bladder axon counts in aging-associated DU [8]. Only a few studies have described ultrastructural defects in the DU bladder; a comprehensive morphological investigation of the entire bladder is still lacking. In addition, previous studies of DU only focused on a single etiology, and the differences among various possible etiologies of DU are unclear. Furthermore, the association between ultrastructural morphological changes and clinical parameters has not been investigated. Currently, pharmaceutical treatment for improving detrusor function in patients with DU is still not available. The understanding of bladders with DU of various etiologies has the potential to promote new drugs for improving detrusor function. Hence, the present study primarily aimed to comprehensively investigate ultrastructural changes in human bladders with DU of various etiologies, including analyses of the epithelium, lamina propria, and muscle layers, using EM. Secondarily, the differences in ultrastructural changes among various DU etiologies, their correlation to urodynamic parameters, and their impact on clinical treatment outcomes were determined.

## 2. Material and Methods

### 2.1. Patient Selection and Bladder Specimens

Patients with DU who were admitted to our hospital from 2019 to 2021 for surgical treatment were prospectively enrolled in the study. A clinical diagnosis of DU was made according to the International Continence Society terminology definition and was confirmed by VUDS [1]. The surgical treatments for patients with DU included endoscopic surgery to eliminate bladder outlet resistance and facilitate voiding (transurethral resection/incision of prostate or bladder neck and urethral sphincter botulinum toxin injection) and intravesical botulinum toxin injection for concurrent detrusor overactivity. Patients with concurrent urological diseases, such as acute or chronic bacterial cystitis, urolithiasis, radiation-related cystitis, or interstitial cystitis were excluded. The institutional review board and ethics committee of Buddhist Tzu Chi General Hospital approved this study (IRB number: IRB108-45-A, IRB108-187-A). All participants provided informed consent before enrollment. Age-matched patients with stress urinary incontinence who were admitted to the hospital for anti-incontinence surgery were enrolled as control subjects. Endoscopic random cold-cup biopsies of the posterior bladder wall were performed in the patients with DU during surgery. Posterior-wall bladder biopsies were obtained from the control subjects during anti-incontinence surgery [9,10]. Each specimen was at least 2 mm in diameter, and endoscopic electrocauterization was performed to prevent bladder bleeding. The specimens were collected from the bladder mucosa, submucosa, and possible muscle layers.

### 2.2. Evaluation of Clinical Parameters

All patients underwent comprehensive medical history reviews after admission. Video urodynamic study (VUDS) was performed for all patients, and detrusor voiding pressure (Pdet), Qmax (maximum flow rate), first sensation of filling (FSF), fullness sensation (FS), urge sensation (US), cystometric bladder capacity (CBC), voided volume, and post-voiding residual urine volume (PVR) were recorded. The patients with DU were classified into three groups according to their medical history and urodynamics results: bladder outlet obstruction (BOO, patients with obviously enlarged prostate volume >40 mL and a tight bladder neck or urethral sphincter on VUDS), neurogenic group (history of central or peripheral neuropathy), or idiopathic group. The treatment outcomes were evaluated 3 months postoperatively. Patients who regained self-voiding and no longer needed urethral catheterization were considered as having successful outcomes, whereas those with a large PVR (voiding volume/functional bladder capacity > 50%) and those who needed urethral catheterization were considered as having unsuccessful outcomes.

### 2.3. Transmission Electron Microscopy

All bladder specimens were analyzed by transmission electron microscopy using a Hitachi H-7500 transmission electron microscope. The procedure of specimen handling for transmission electron microscopy was described in our previous study [9]. The epithelium (urothelium), lamina propria, and muscle layers were analyzed separately. The specimens obtained from the control subjects were regarded as normative bladders, and the defects or abnormal findings in the DU bladders were graded using a 3-point scale. In the specimens with a detrusor muscle layer, the smooth muscle morphology and intercellular space were evaluated. For the epithelium layers, the specimens with epithelium covered in more than 75% bladder tissue were considered as normal, among 25–75% epithelium was considered as mild denudation, and less than 25% epithelium was considered as total denudation. For the intercellular space, the specimens with all epithelial cells in close contact were considered normal, the intercellular space enlargement noted among 0–50% of epithelial cells was considered mildly enlarged, and more than 50% of the area of the epithelium having enlarged intercellular space was considered as severely enlarged. The bladder specimens with more than 10 tight junctional complexes in the umbrella cells were classified into normal junctional complexes, and the others were broken junctional complexes. For the lamina propria, bladder specimens with large, aggravated lymphocytes were considered as severe lymphocyte infiltration, specimens with significant lymphocytes distribution were considered as mild, and the other bladder specimens were considered as normal. The bladder specimens with lamina propria collagen deposition in less than 50% space were considered as loose connective tissue density, and the others were considered as normal. The bladder specimens with more than 10 damaged submucosa vessels were considered as broken, the bladder specimens with more than 5 identifiable nerves in lamina propria were considered as submucosa nerve terminal presented, and the bladder specimens with more than 5 identifiable telocytes in lamina propria were considered as telocytes presented. The bladder specimens with detrusor muscle cells shrinkage, enlarged intercellular spaces and irregular contour were considered as shrinkage smooth muscle cells. Four representative specimens of bladder muscle cells were taken from each sample for quantification. Using ImageJ software, the muscle cell border was outlined and the muscle cell area was measured. The proportions of muscle cells in the specimens were calculated. All of the EM results were investigated and graded by a single investigator, Dr. Han-Chen Ho, who was masked to the clinical results.

### 2.4. Statistical Analysis

The patients with DU were classified according to the possible etiology or treatment outcome, and the chi-square test was used to analyze the significance of differences in EM characteristics between the DU groups. The patients with DU were categorized into two or three groups according to the severity of the urothelial defects on EM. Differences in the quantitative urodynamics of symptom parameters between the EM finding groups were compared using a non-parametric test. *p*-values of < 0.05 were considered significant. All statistical analyses were performed using SPSS for Windows, version 16.0 (SPSS, Chicago, IL, USA).

## 3. Results

A total of 25 patients with DU (16 men and 9 women) were enrolled and provided bladder specimens for EM investigation, and 4 control subjects were enrolled. The mean ages of the patients with DU and control subjects were 74.2 ± 10.9 and 72.1 ± 6.8 years, respectively. The mean history of voiding problems among the patients with DU was 4.75 ± 2.30 years. The urodynamic results are shown in Table 1. Based on the etiology, 12 patients were classified as BOO, and 6 and 7 patients were classified as neurogenic and idiopathic, respectively. Eighteen patients had successful treatment outcomes, whereas seven patients had unsuccessful outcomes. Patients with successful treatment outcomes had significantly smaller US and CBC at baseline and a higher proportion of BOO etiology (Table 1). Age was not associated with the treatment outcome.

Compared with the normative bladders from control subjects, DU bladders exhibited multiple ultrastructural defects in the epithelium, lamina propria, and muscle layers on EM. Table 2 details the abnormal EM findings in the DU bladders. Figure 1 shows representative images of various grades of epithelial cell layer defects, including an enlarged intercellular space, umbrella cell uroplakins, and tight junction complex defects in the epithelium of DU bladders. Six (24%) patients with DU had complete denudation of epithelial cells, whereas twelve patients with DU had intact normal epithelial cell layers. An enlarged intercellular space was noted in 83.3% of 19 patients with preserved epithelial cells. The tight junction complexes in the umbrella cells were broken in 44.4 % of patients with DU, and only 22.2% of patients with DU had intact uroplakins in the umbrella cells. None of the abovementioned epithelial defects were noted in the four control bladders, although one had loose epithelial cells.

On EM, DU lamina propria specimens exhibited various grades of loose extracellular connective tissue density, lymphocyte infiltration, and broken blood vessels (Table 2 and Figure 2). Lamina propria nerve terminals were only found in 6 of the 25 (24%) DU bladders (Figure 2H), whereas they were observed in 3 of the 4 (75%) control bladders. Telocytes were only found in two DU bladders (Figure 2I). Detrusor muscles were noted in 12 of the 25 DU bladders and all control bladders. The control detrusor muscle cells in EM showed in the Figure 3A,B. The detrusor muscle cells in the DU bladders were characterized by a loss of regular spindle shape in longitudinal sections or round shape in cross-sections and became irregular (Figure 3C,D). A ruffled cell contour with several cytoplasmic processes spanning the enlarged intercellular space was evident. Smooth muscle shrinkage and enlarged intercellular spaces were present in 11 of the 12 (91.6%) DU bladders with detectable detrusor layer (Table 1) but not in the control bladders. The median proportion of muscle cell area in the DU bladders was 49% (range, 30–62%), whereas that in the control bladders was 68% (range, 68–79%; *p* < 0.001) (Figure 3E).

An intact epithelial layer and the presence of lamina propria nerve terminals were significantly correlated with the urodynamic parameters. Patients with DU with intact epithelial cell layers had significantly larger voided volumes and Qmax than those with mild or severe epithelial denudation (Table 3). The patients with DU with nerve terminals in the lamina propria had smaller FSF, FS, and PVR than the patients without nerve terminals in the lamina propria. The other bladder EM findings were not correlated with the urodynamic parameters (Appendix A). The treatment outcomes and etiologies of DU were not associated with the bladder findings on EM. Age did not differ significantly between the patients with DU and the various EM findings.

## 4. Discussion

DU is a common urological problem, however, it has attracted limited research attention. While several recent studies have investigated the changes in function receptors in human DU bladders [3], the anatomical defects in human DU bladders have not been thoroughly examined. The present study investigated DU of various etiologies using EM to clarify morphological abnormalities in the epithelium, lamina propria, and muscle layers. Generally, most DU bladders exhibited epithelial denudation, large epithelial intercellular spaces, lack of uroplakins, and tight junction complexes in the umbrella cells. The lamina propria was characterized by loose connective tissue, broken vessels, and a lack of nerve terminals. Smooth muscle shrinkage was also noted in the detrusor layer of DU bladders. Intact epithelial layers and the presence of lamina propria nerve terminals were associated with urodynamic parameters. The morphological findings of bladders on EM were not different between the patients with DU of various etiologies and treatment outcomes.

The ultrastructural morphological changes in the detrusor muscle have been widely investigated in the field of functional urology. A previous EM study of bladders from patients with BOO-associated detrusor overactivity showed hypertrophic detrusor muscle cells with markedly widened intercellular spaces [11]. The widened intercellular spaces in the bladder detrusor layer were also noted in the present study, but the DU detrusor was further characterized by shrunken muscle cells with ruffled contours. These findings affirm the well-known pathogenetic mechanism in which BOO induces bladder muscle hypertrophy with overactivity and then progresses to the decompensation stage with loss of functional emptying ability [12]. The detrusor muscle cells in BOO bladders first become hypertrophic and then progress to muscle shrinkage, leading to detrusor muscle decompensation. Notably, detrusor muscle cell shrinkage was also observed in both neurogenic and idiopathic DU, suggesting that the pathogenesis of neurogenic or idiopathic DU also involves detrusor muscle failure.

The role of epithelial cells and morphological changes in the lamina propria in DU bladders is not well investigated in the literature. Previous EM studies on patients with interstitial cystitis/bladder pain syndrome (IC/BPS) revealed a decreased thickness of bladder epithelial cell layers, widened intercellular spaces, and microvilli in the umbrella cells [9,13]. Loss of tight junction complexes and uroplakins in the umbrella cells in the bladder have also been noted in detrusor overactivity [14,15]. Our study also found similar ultrastructural findings of the bladder in patients with DU, suggesting that such features may not be specific to IC/BPS, overactive bladder, or DU. Decreased thickness of epithelial cell layers, large intercellular spaces, and the loss of tight junction complexes in the apical urothelial cells indicate immature uroepithelial cells and might be common characteristics of an unhealthy bladder. In the present study, patients with DU with normal bladder epithelial cell layers had larger voided volumes and Qmax. An intact bladder epithelium in patients with DU indicates a relatively healthier bladder function and better sensation or neural coordination to the empty bladder with abdominal straining.

Suburothelial innervation has been proven to play an important role in both bladder sensation and detrusor contractility [16,17]. Some patients with DU had atonic bladders that lacked any sensation. In contrast, some patients with DU were characterized as having increased bladder sensation and even urinary incontinence [18]. Most patients with DU in this study had loose connective tissue in the lamina propria and lacked lamina propria nerve terminals. The patients with suburothelial nerves had better bladder sensation in the urodynamic study and smaller PVR than those without suburothelial nerves. The presence of nerves in the lamina propria indicated preserved bladder sensation in patients with DU, which might be important for abdominal straining to empty the bladder. Similar to detrusor muscle cell shrinkage, the absence of lamina propria innervation was also common among patients with DU of different etiologies. Suburothelial nerve deficits are also involved in the pathogenetic mechanism in idiopathic, neurogenic, and BOO DU.

The current study showed that bladder ultrastructural findings were not significantly different between the DU patients with different etiologies. Evidence from early studies revealed nerve injury might induce muscle atrophy [19] so that the detrusor muscle atrophy might be a reasonable bladder change in patients with neurogenic DU. In the BOO-associated bladder disorder, ischemia was believed to play an essential role in the pathogenesis mechanism [3]. Bladder ischemia may not only cause detrusor muscle cell shrinkage but also might directly result in nerve degeneration or absence in the bladders [3]. Our study showed that the treatment outcome was not significantly different between the DU patients with different EM findings. Since our treatment mainly focused on relief of bladder outlet resistance and the patients’ need to void with abdominal straining, bladder function, including sensory or detrusor function, may not be the essential factor in determining the treatment outcome.

The main limitation of this study is the small number of cases in each DU group. Most EM findings in the study were only analyzed by semi-quantification grading, and the subjective grading bias might have affected the results. Specimens from the bladder posterior wall may not be representative of the entire bladder; thus, the pathological changes reported here may not be homogeneous throughout the bladder. The current study is a cross-section observational study, and we only enrolled the DU patients with long-term severe voiding problems and requiring surgical intervention. Hence, the results of this study only include the DU patients in late-stages. The progressive DU bladder changes have not been investigated yet. Co-morbidities such as diabetes in the patients with DU might also cause morphological bladder changes; however, the impact of the co-morbidities on our result has not been investigated in this study. This study was also limited by only a single investigator to analyze the bladder ultrastructural changes. Although age did not vary significantly among the patients with DU with various EM findings, it might still impact ultrastructural changes in the bladder. The treatment outcomes in the patients with DU were determined by bladder-emptying function, but the bladder-emptying ability might have resulted from abdominal straining rather than from detrusor contractility recovery.

## 5. Conclusions

The human DU bladders were characterized by ultrastructural defects in the entire bladder, including loss of uroplakins and tight junction complexes in apical epithelial cells, looser connective tissue and the absence of nerve terminals in the lamina propria, and detrusor muscle cell shrinkage and loss of spindle shape. The bladder urothelium denudation and absence of nerve terminals in EM were negatively correlated with the urodynamic parameters. These defects were observed in DU bladders of various etiologies. DU, due to various etiologies, may have different mechanisms but share similar ultrastructural pathological changes.

## Figures and Tables

**Figure 1 biomedicines-10-01055-f001:**
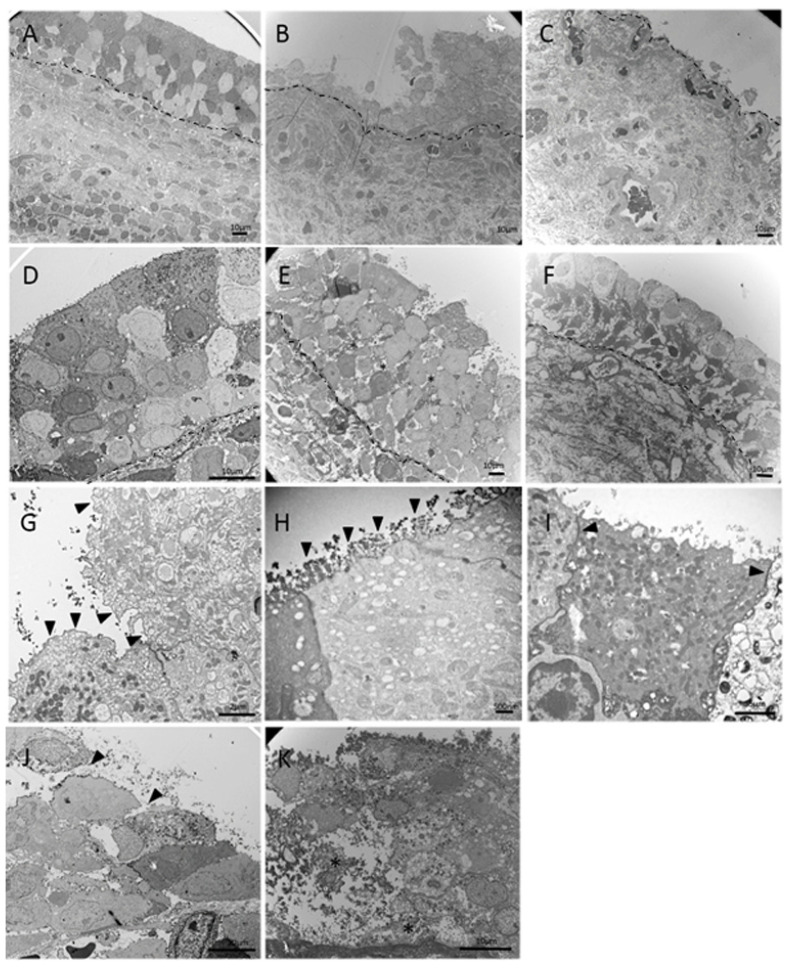
Representative images of various grades of EM bladder epithelial cell defects in the patients with DU. (**A**) Normal intact epithelium with 3–7 cell layers; (**B**) mild epithelial denudation; (**C**) completely denudated epithelium; (**D**) normal epithelial intercellular space; (**E**,**F**) mild and severe enlarged epithelial intercellular spaces (indicated by asterisks, *); (**G**) apical umbrella cell uroplakins (black triangle); (**H**) apical umbrella cells with microvilli (black triangle); (**I**) epithelial cells with intact tight junction complexes (black triangle); (**J**) epithelial cells without tight junction complexes; and (**K**) lysed epithelium cells. Dotted lines indicate the basement membrane.

**Figure 2 biomedicines-10-01055-f002:**
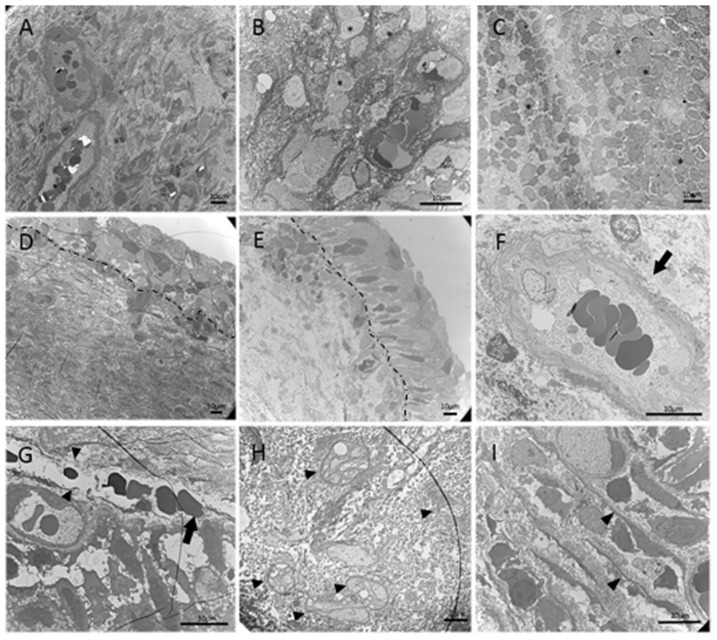
Representative figures for various grades of EM bladder lamina propria defects in patients with DU. (**A**) Normal lamina propria without lymphocyte infiltration; (**B**,**C**) mild and severe lamina propria lymphocyte infiltration (lymphocytes indicated by asterisks, *); (**D**) normal lamina propria extracellular connective tissue density; (**E**) loose lamina propria extracellular connective tissue; (**F**) intact lamina propria vessels (arrow); (**G**) broken lamina propria vessels (black triangle) with red blood cells extravasation (arrow); (**H**) presence of nerve terminals in der lamina propria (black triangle); (**I**) presence of telocytes in the lamina propria (black triangle). Dotted lines indicate the basement membrane.

**Figure 3 biomedicines-10-01055-f003:**
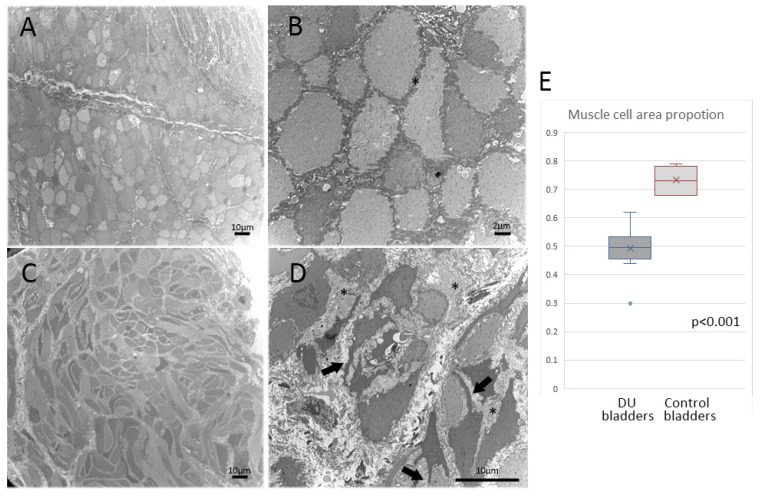
Representative images of EM detrusor muscles. (**A**,**B**) Bladder detrusor layer images taken from control subjects showing normal smooth muscle morphology with tight intercellular spaces (indicated by asterisks, *). (**C**,**D**) Bladder detrusor layer images taken from patients with DU showing loss of regular spindle shape in longitudinal sections or round shape in cross-sections. Ruffled cell contour with several cytoplasmic processes (arrows) spanning the enlarged intercellular space (*). (**E**) Median proportion of muscle cell area in DU (*n* = 12) and control bladders (*n* = 4).

**Table 1 biomedicines-10-01055-t001:** Clinical parameters of patients with DU with different treatment outcomes.

	Total Patients (*n* = 25)	Successful	Unsuccessful	*p*-Value
Age	75.2 ± 13.4	74.4	77.3	0.569
Sex	16 male/9 female			
FSF (mL)	234.2 ± 118.9	214.9	280.9	0.225
FS (mL)	319.6 ± 114.9	314.1	358.6	0.415
US (mL)	383.6 ± 159.2	336.0	499.3	0.019
Pdet (cm H_2_O)	12.3 ± 15.3	15.1	5.57	0.065
Qmax (mL/s)	1.6 ± 2.6	1.76	1.29	0.689
Voided Volume (mL)	23.2 ± 34.0	22.8	24.1	0.933
PVR (mL)	440.5 ± 210.8	392.5	575.0	0.084
CBC (mL)	459.8 ± 210.6	406.9	586.3	0.045
Etiology				
Idiopathic	7	5	2	0.034
neurogenic	6	2	4	
BOO	12	11	1	

FSF: first sensation of filling; FS: fullness sensation; US: volume of urge sensation; Pdet: detrusor voiding pressure; Qmax: maximal flow rate; PVR: post-voiding residual urine volume; CBC: cystometric bladder capacity; BOO: bladder outlet obstruction. The treatments for patients with DU included transurethral resection/incision of prostate or bladder neck and urethral sphincter botulinum toxin injection.

**Table 2 biomedicines-10-01055-t002:** Proportion of EM bladder defects in patients with DU.

**Epithelium EM Findings**		Normal	Mild Denudated	Total Denudated
	Epithelium layers	48.0%	28.0%	24.0%
		Normal	Mid enlarged	Severe enlarged
	Intercellular space	16.7%	50.0%	33.3%
		Normal	broken	
	Tight junctional complex	55.6%	44.4%	
		uroplakin	microvilli	
	Apical cell morphology	22.2%	77.8%	
**Lamina Propria EM Findings**		Normal	Mild	Severe
	Lymphocyte infiltration	64%	20%	16%
		Normal	loose	
	Connective tissue density	52.0%	48.0%	
		Normal	broken	
	Submucosa vessels	64.0%	36.0%	
		presented	absence	
	Submucosa nerve terminal	24.0%	76.0%	
		presented	absence	
	telocytes	8.0%	92.0%	
**Detrusor EM Findings**		Normal	shrinkage	
	Smooth muscle morphology	8.4%	91.6%	

The percentage indicated the proportion of each EM characteristic classification among the DU patients.

**Table 3 biomedicines-10-01055-t003:** Clinical parameters in patients with DU with bladder EM defects of varying severities.

	Normal Epithelium Layer *n* = 12	Mild Denudated Epithelium *n* = 7	Severe Denudated Epithelium *n* = 6	*p*-Value *	Nerve Presence *n* = 6	Nerve Not Presence *n* = 19	*p*-Value ^#^
age	74.6 ± 13.7	78.7 ± 5.5	72.5 ± 9.5	0.586	71.2 ± 18.9	76.5 ± 7.2	0.524
FSF (mL)	234 ± 116	244 ± 133	214 ± 131	0.899	152 ± 65	262 ± 121	0.047
FS (mL)	328 ± 114	346 ± 127	305 ± 135	0.843	223 ± 102	362 ± 104	0.009
US	344 ± 117	403.5 ± 123.9	443.5 ± 250.6	0.448	284.2 ± 137.1	416.8 ± 155.3	0.077
Pdet (cmH_2_O)	9.3 ± 13.1	17.8 ± 20.2	12.8 ± 15.2	0.557	12.5 ± 17.5	12.3 ± 15.0	0.976
Qmax (mL/s)	2.9 ± 3.1	0.3 ± 0.8	0.3 ± 0.8	0.042	1.33 ± 1.21	1.72 ± 2.93	0.757
Voided Volume (mL)	44.3 ± 37.7	2.0 ± 4.9	2.3 ± 5.72	0.005	21.2 ± 28.1	23.9 ± 36.4	0.869
PVR (mL)	385.8 ± 197.4	460.0 ± 114.0	560.0 ± 328.6	0.342	305.0 ± 123.9	493.8 ± 229.4	0.025
CBC (mL)	430.1 ± 203.6	460.3 ± 103.2	519.0 ± 309.7	0.718	326.2 ± 121.6	504.4 ± 217.4	0.071

FSF: first sensation of filling; FS: fullness sensation; US: volume of urge sensation; Pdet: detrusor voiding pressure; Qmax: maximal flow rate; PVR: post-voiding residual urine volume; CBC: cystometric bladder capacity; *: The *p*-value indicated the result of the one-way ANOVA test for the DU patients with normal, mild, and severe denudated epithelium; #: The *p*-value indicated the result of the independent t-test for the DU patients with nerve presence in lamina propria or not.

## Data Availability

Data are available by contacting the corresponding authors.

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
