# Peer review of "Clinical Correlation of Bladder Electron Microscopic Characteristics in Patients with Detrusor Underactivity of Various Etiologies"

_biomedicines, 2022, doi:10.3390/biomedicines10051055_

Round 1
Reviewer 1 Report
The manuscript aims to describe the ultrastructural characteristics of the bladder of patients 12 with detrusor underactivity (DU) of various etiologies. The study is well-written however, as the authors correctly reported the main limitation is linked to the low number of patients that should be enlarged. Indeed, it is not a rare pathology and a definite conclusion may not be reached with 25 cases.
Please add how many pathologists participate in the study and what are the experiences.
Author Response
The manuscript aims to describe the ultrastructural characteristics of the bladder of patients 12 with detrusor underactivity (DU) of various etiologies. The study is well-written however, as the authors correctly reported the main limitation is linked to the low number of patients that should be enlarged. Indeed, it is not a rare pathology and a definite conclusion may not be reached with 25 cases.
Please add how many pathologists participate in the study and what are the experiences.
Reply: Thanks for your comment and question. Indeed, the small case number and the experiences of the pathologist is the main limitation of this study. Because bladder biopsy is not a routine procedure for patients with DU, most urologists and pathologists did not have the experiences to investigate what is the pathological changes in the bladders of patients with DU. The current study focused on the DU bladder ultrastructural changes in electron microscopic (EM), and all of the EM results were investigated and graded by a single investigator Dr. Han-Chen Ho who was masked to the clinical results. Dr. Ho is a human anatomist and an expert on EM. She previously had published a study to investigate the bladder EM changes in patients with interstitial cystitis (PLoS One. 2018, 7;13:e0198816). This study was also indeed limited by an only single investigator to analyze the bladder changes. We added these into the manuscript (page 3, line 133 to 134).
Reviewer 2 Report
This submission investigated the patients with detrusor underactivity (DU) to compare with control subjects through urodynamic testing and transmission electron microscopic examination of bladder specimens. Please conduct the concerns below.
- The morphological and anatomic studies of DU bladders were limited. Why?
- Merits of understanding in bladders with DU of various etiologies were not introduced, particularly in clinical practice.
- Posterior-wall bladder biopsies were obtained from the control subjects during anti-incontinence surgery. Please add the reference(s) to support.
- Quantification of the changes in histological picture needs to introduce in detail.
- In Table 1, DU with different treatment seems unclear. Please add in legends. Additionally, p value was not indicated in last two factors.
- Table 2 needs to explain the percentage of what?
- Table 3 seems hard to know the p value for difference between three parameters. Additionally, sample size for each value remained unclear.
- The morphological findings in bladders by EM were not different between the patients with DU of various etiologies and treatment outcomes. Please discuss it in clear. What is the treatment? What is the potential mechanism for this result?
- Progress of the changes in patients with DU seems ignored. Additionally, the combined disorder(s) in outcomes were not included.
- Novelty of current study was not indicated in conclusion.
Author Response
This submission investigated the patients with detrusor underactivity (DU) to compare with control subjects through urodynamic testing and transmission electron microscopic examination of bladder specimens. Please conduct the concerns below.
1. The morphological and anatomic studies of DU bladders were limited. Why
Reply: Thanks for your question. The morphological and anatomic studies should be the fundamental to pathogenesis research. However, in comparison to molecular and physiological, the results of the morphological study might be more difficult in quantification without subjective evaluation. For analyzing the results of the morphological bladder study, the researchers must have experience with the human bladder ultrastructure. With the advantage of obtaining human bladder specimens, our research team previously had published several articles to investigate the human bladder ultrastructure (PLoS One. 2018; 13(6): e0198816; Sci Rep. 2021; 11: 17258; Tzu Chi Med J. 2020,16;33(4):345-349).
2. Merits of understanding in bladders with DU of various etiologies were not introduced, particularly in clinical practice.
Reply: Thanks for your comments. Currently, pharmaceutical treatment for improving detrusor function in patients with DU is still not available. Hence, the understanding of bladders with DU of various etiologies has potential to promote new drugs for improving detrusor function. We added above explanation to the introduction section. (page 2, line 59 to 61).
3. Posterior-wall bladder biopsies were obtained from the control subjects during anti-incontinence surgery. Please add the reference(s) to support.
Reply: Thanks for your question. We enrolled patients with pure stress urinary incontinence without bladder abnormality in the urodynamic study as the control subjects. Patients with SUI are believed to be the good control subjects to provide bladder specimens for functional urology investigation. We added the reference to the method section (page 2, line 84).
4. Quantification of the changes in histological picture needs to introduce in detail.
Reply: Thanks for your question. For the epithelium layers, the specimens with epithelium covered in more than 75% bladder tissue were considered as normal, among 75-25% epithelium were considered as mild denudation, and less than 25% epithelium was considered as total denudation. For the intercellular space, the specimens with all epithelial cells contact closely were considered as normal, the intercellular space enlargement noted in among 100-50% of epithelial cells was considered mildly enlarged, and more than 50% area of the epithelium having enlarged intercellular space was considered as severe enlarged. The bladder specimens with more than 10 tight junctional complexes in the umbrella cells were classified into normal junctional complexes, and the others were broken junctional complexes. For the lamina propria, bladder specimens with large aggravated lymphocytes were considered as severe lymphocyte infiltration, specimens with significant lymphocytes distribution were considered as mild, and the other bladder specimens were considered as normal. The bladder specimens with lamina propria collagen deposition in less than 50% space were considered as loose connective tissue density, and the others were considered as normal. The bladder specimens with more than 10 damaged submucosa vessels were considered as broken, the bladder specimens with more than 5 identifiable nerves in lamina propria were considered as submucosa nerve terminal presented, and the bladder specimens with more than 5 identifiable telocytes in lamina propria were considered as telocytes presented. The bladder specimens with detrusor muscle cells shrinkage, enlarged intercellular spaces and irregular contour were considered as shrinkage smooth muscle cells. We added these sentence to the method section (page 3, line 110 to 130).
5. In Table 1, DU with different treatment seems unclear. Please add in legends. Additionally, p value was not indicated in last two factors.
Reply: Thanks for your question. The treatments included transurethral resection/incision of prostate or bladder neck and urethral sphincter botulinum toxin injection, we added above mentioned methods into the table legends (page 4, line 158 to line 159). The p-value in the last column of table 1 was the chi-square test result which compares the treatment outcome in patients with different etiologies (Idiopathic, neurogenic, BOO), so there is only one p-value.
6. Table 2 needs to explain the percentage of what?
Reply: Thanks for your question. The percentage in the table indicated the proportion of each EM characteristic classification among the DU patients. We added these into the table legend (page 5, line 173).
7. Table 3 seems hard to know the p value for difference between three parameters. Additionally, sample size for each value remained unclear.
Reply: Thanks for your suggestion. We added the sample number in each groups and the explanation for the p-value in table 3 (page 8, line 232 to page 9, line 236).
8. The morphological findings in bladders by EM were not different between the patients with DU of various etiologies and treatment outcomes. Please discuss it in clear. What is the treatment? What is the potential mechanism for this result?
Reply: Thanks for your suggestion. Current study showed that bladder ultrastructural findings were not significantly different between the DU patients with different etiologies. Evidence from early studies revealed nerve injury might induce muscle atrophy so that the detrusor muscle atrophy might be a reasonable bladder change in the patients with neurogenic DU. In the BOO-associated bladder disorder, ischemia was believed to play an essential role in the pathogenesis mechanism. The bladder ischemia may not only cause detrusor muscle cells shrinkage but also might directly result in nerve degeneration or absence in the bladders. Our study showed that the treatment outcome was not significantly different between the DU patients with different EM findings. Since our treatment mainly focused on relief of bladder outlet resistance and the patients need to void with abdominal straining, bladder function, including sensory or detrusor function, may not be the essential factor in determining the treatment outcome. We added above discussion into the manuscript (page 10, line 289 to line 300).
9. Progress of the changes in patients with DU seems ignored. Additionally, the combined disorder(s) in outcomes were not included.
Reply: Thanks for your comment. Indeed, the current study is a cross-section observational study, and we only enrolled the DU patients with long-term severe voiding problem (mean history of voiding problem was 4.75±2.30 years) and requiring surgical intervention. Hence, the results of this study only include the DU patient in late-stage. The progressive DU bladder changes have not been investigated yet. The co-morbidities such as diabetes in the patients with DU also might cause morphological bladder changes, but the impact of the co-morbidities on our result has not been investigated in this study. We added these into the limitation and results sections (page 4, line 148 to 149; and page 10, line 305 to 312)
10. Novelty of current study was not indicated in conclusion.
Reply: Thanks for your comment. This study is the first human study to comprehensively investigate the entire bladder ultrastructural changes in patients with DU. We not only observed the bladder changes in patients with DU, but also found the morphological changes were associated with the clinical urodynamic parameters. We added this point into the conclusion section (page 10, line 322 to line 324).
Round 2
Reviewer 1 Report
None.
Reviewer 2 Report
It has been revised according to comments in a good way.